# Integrative Effects between a Bubble and Seal Program and Workers’ Compliance to Health Advice on Successful COVID-19 Transmission Control in a Factory in Southern Thailand

**DOI:** 10.3390/ijerph192416391

**Published:** 2022-12-07

**Authors:** Chanon Kongkamol, Thammasin Ingviya, Sarunyou Chusri, Smonrapat Surasombatpattana, Atichart Kwanyuang, Sitthichok Chaichulee, Intouch Sophark, Chaiwat Seesong, Thanawan Sorntavorn, Tanyawan Detpreechakul, Pindanunant Phaiboonpornpong, Kamol Krainara, Pornchai Sathirapanya, Chutarat Sathirapanya

**Affiliations:** 1Department of Family and Preventive Medicine, Faculty of Medicine, Prince of Songkla University, Hat Yai 90110, Songkhla, Thailand; 2Air Pollution and Health Effect Research Center, Prince of Songkla University, Hat Yai 90110, Songkhla, Thailand; 3Department of Internal Medicine, Faculty of Medicine, Prince of Songkla University, Hat Yai 90110, Songkhla, Thailand; 4Department of Pathology, Faculty of Medicine, Prince of Songkla University, Hat Yai 90110, Songkhla, Thailand; 5Department of Biomedical Sciences and Biomedical Engineering, Faculty of Medicine, Prince of Songkla University, Hat Yai 90110, Songkhla, Thailand

**Keywords:** COVID-19, factory, transmission, quarantine, dormitory

## Abstract

Applying health measures to prevent COVID-19 transmission caused disruption of businesses. A practical plan to balance public health and business sustainability during the pandemic was needed. Herein, we describe a “Bubble and Seal” (B&S) program implemented in a frozen seafood factory in southern Thailand. We enrolled 1539 workers who lived in the factory dormitories. First, the workers who had a high fatality risk were triaged by RT-PCR tests, quarantined and treated if they had COVID-19. Newly diagnosed or suspected COVID-19 workers underwent the same practices. The non-quarantined workers were regulated to work and live in their groups without contact across the groups. Workers’ personal hygiene and preventive measures were strongly stressed. Between the 6th and 9th weeks of the program, the post-COVID-19 infection status (PCIS) of all participants was evaluated by mass COVID-19 antibody or RT-PCR tests. Finally, 91.8% of the workers showed positive PCIS, which was above the number required for program exit. Although no workers had received a vaccination, there was only one case of severe COVID-19 pneumonia, and no evidence of COVID-19 spreading to the surrounding communities. Implementation of the B&S program and workers’ adherence to health advice was the key to this success.

## 1. Introduction

Severe acute respiratory syndrome coronavirus-2 (SARS-CoV2) or novel Coronavirus-2 disease, currently named as Coronavirus disease-19 (COVID-19), is a rapidly transmissible severe acute respiratory infection that caused both health and economic losses globally. After the first outbreak of COVID-19 in Wuhan, China in December 2019, it was declared a global health crisis by the WHO in February 2020. In Thailand, three COVID-19 waves occurred in the years 2019, 2020 and 2021. Among these, the second wave originated in a shrimp-selling central market in Samutsakorn province in central Thailand. Because of the rush and busy commercial activities in the market, overcrowded living conditions of the migrant labor workers in the market, and possibly loose compliance to COVID-19 transmission prevention regulations issued by the Ministry of Public Health of Thailand (MOPH-T), the number of COVID-19 cases among the migrant workers of the market as well as people in the surrounding communities rapidly increased. This necessitated activation of a mass screening program for COVID-19 and quarantining of individuals who were suspected to have contracted the disease [1]. The migrant workers were the largest portion of the persons involved in the activities of the market and considered to be the originating center of this outbreak. These migrant workers had immigrated for labor work from Thailand’s neighboring countries. The outbreak in Samutsakorn marked a notable change in the characteristics of COVID-19 epicenters of outbreaks, from night entertainment venues such as bars, night clubs and discotheques, as before, to markets, factories and construction worker camps. Hence, this outbreak led the government to activate mass COVID-19 screenings among the worker communities around Thailand, such as at construction sites, fresh food markets, factories, etc. Migrant workers have been an important manpower source in many enterprises around Thailand, including in Songkhla province in the south of the country, and mass travelling of these workers was thought to be a significant factor in the rapid transmission of COVID-19 during the epidemic. A study conducted in China showed that many dense industrial or commercial cities located near COVID-19 epicenters or serving as transportation junctions had significantly higher risks of becoming emerging COVID-19 epicenters [2]. In addition, the overcrowded living or working conditions of these workers, as well as their rapid turnover, facilitated the rapid transmission of COVID-19, as had happened previously with other respiratory tract diseases [3,4,5,6]. Particularly, workers with asymptomatic COVID-19 could unknowingly transmit the disease to others. Based on the conditions mentioned, the migrant workers who resided in factory dormitories or worker camps were highly vulnerable to both contracting and transmitting COVID-19 in their workplaces and subsequently to surrounding communities. [7]. To reduce the COVID-19 transmission rate, several personal hygiene and preventive measures were introduced by the MOPH-T. One of the measures, known as “DMHTT”, which stands for (physical) distancing, wearing masks, hand hygiene, temperature checks and COVID-19 antigen tests, was strongly encouraged.

During the time of the third COVID-19 wave in Thailand, several social gathering places or workplaces were ordered to completely lock down because of fears of viral transmission. Many shops, social activity venues and factories in Thailand were required to close with the aim of lowering the COVID-19 infection rate, as in many other countries [8,9,10]. Apart from the factory lockdowns, one study reported that absolute locking of the workers in their dormitories was effective in lowering the infection rates [11]. However, such strict lockdown measures resulted in unprecedented economic losses globally. Planning measures to balance disease transmission control while allowing the sustainability of an enterprise during the epidemic was challenging for public health and industrial policy makers. In August 2021, the MOPH-T launched the “Bubble and Seal” program (B&S) to attempt to balance these contradictory objectives [12]. The principle of this program was that, while the measures for COVID-19 transmission control would be strictly applied, the regular industrial or business activities could continue without significant disruption. Practically, grouping the workers working or living in the same workplaces or dormitory rooms (bubbles) and prohibiting them from having contact with other groups or people outside their worksites or dormitory rooms (seal) were the key actions of the program. This program aimed to contain the COVID-19 infection in the workplace completely until the percentage of post-COVID-19 infection workers (PCIW), as confirmed by positive antigen (recent) or antibody tests (remote), plus the percentage of COVID-19 vaccinated workers, achieved the number indicated by the MOPH-T (>85%) so that the program could be ended.

Frozen seafood factories comprise the major number of factories in Songkhla province and migrant workers from neighboring countries are the main manpower hired in these factories. When the rate of COVID-19 among the migrant workers of these factories increased rapidly with a high possibility of spreading to the public, the Songkhla Provincial Public Health Office (SPPHO) implemented a B&S program in cooperation with our faculty COVID-19 control team and the factory owners. Herein, we describe the progression of the COVID-19 situation in a frozen seafood factory in Songkhla, the implementation of the B&S program, the outcomes of the program and the impact on the factory’s surrounding communities.

## 2. Materials and Methods

### 2.1. Study Participants and Setting

There were 2956 workers in this factory, composed of 1634 (55.3%) migrant workers and 1322 local Thai workers. Among these, 1539 of the migrant workers resided in the factory dormitories (live-in workers, LIW) and were enrolled as study participants. The rest of the workers lived in their homes around the factory (live-out workers, LOW). We focused on the LIW as they were considered the more vulnerable group for acquiring and transmitting COVID-19 than the LOW because of their crowded living conditions in the factory dormitories and their socializing habits. The industrial product of the factory were frozen transformed seafoods sealed in vacuum packs. Human power combined with industrial machines was used in the production process. The workers were adequately informed about the necessity, significance and details of their compliance required according to the protocol to control COVID-19. The factory health team emphasized to the workers that the protocol primarily aimed to prevent the workers themselves and the people in the surrounding communities of the factory from contracting the disease and becoming seriously ill. After the information was provided, the workers’ consent was obtained.

### 2.2. Application and Termination of Bubbles and Seal Program

#### 2.2.1. Screening for High Mortality Risk Workers, Quarantining and Modifying Working Conditions

When the number of workers whose real time polymerase chain reaction (RT-PCR) tests confirmed COVID-19 was rapidly increasing in the factory, the SPPHO in cooperation with our faculty staff and the factory owner initiated a B&S program. Following the protocol of the B&S program, we first triaged the LIW who had underlying diseases or medical conditions which made them high risk for COVID-19-related mortality, e.g., high body mass index (BMI), chronic airway diseases, cardiovascular diseases, diabetes mellitus, cancer, or pregnant women, to undergo RT-PCR testing [12]. Those whose RT-PCR tests were positive or who had respiratory symptoms albeit negative RT-PCR tests were quarantined in two pre-specified hospitals for at least 10 days until a repeated RT-PCR test showed negative results before being allowed to return to work as usual. The rest of the workers were allowed to continue their routine work but with strict compliance to the B&S rules and the MOPH practical guidelines for universal prevention of COVID-19 transmission. If a new RT-PCR positive or a suspected infection worker was identified, the case would undergo the same disease control and treatment protocols. We planned to do mass COVID-19 antibody tests (or RT-PCR tests in doubtful diagnosis cases) in all studied LIW to determine the percentage of post COVID-19 infection cases after no new or suspected COVID-19 cases were reported, when it was expected that the COVID-19 infection rate among the LIW had achieved the requirement for program exit.

During the B&S program, all the LIW were provided with a full supply of foods and drinks, medicines, and all necessary material for their daily living in the factory dormitories. At least one translator who knew the workers’ native language was provided to help the communication between the workers and the factory health staff or others. They were not allowed to go out of the factory except for unavoidable necessity or medical reasons and only after receiving the required approval from the factory health team. Meanwhile, the LOW were transported between the factory and their homes by a prespecified, non-stop route in factory-provided pickup trucks, which were modified for prevention of interpersonal COVID-19 transmission. During working hours in the factory, face-to-face interpersonal contacts were limited by using an intercom or telephone and setting up assembly points in the factory for dropping and picking up materials at an overlapping time schedule. All the workers were strongly encouraged to comply with DMHTT measures at all times.

#### 2.2.2. Evaluation the Outcome of B&S Program and Program Exit

After no new cases of COVID-19 were reported in the factory, we randomly selected 133 LIW to undergo COVID-19 antibody tests for a preliminary estimation of the post-COVID-19 infection status in the factory. Thence, the rest of LIW in the factory underwent the antibody tests. Based on the recommended guideline for exiting a B&S program by the National Center of Disease Control of the MOPH-T, the total percentage of people who had received a vaccination, who had a positive RT-PCR and who had a positive COVID-19 antibody test should be >85% to proceed to exiting the program [12]. In our study, we used only the percentage of positive RT-PCR tests (recent infection) or positive COVID-19 antibody tests (remote infection) to define as post-COVID 19 infection status (PCIS), and a modified indicator for exiting the B&S program because no workers in this study had received a COVID-19 vaccine yet. We collectively defined the workers who had positive PCIS as “post-COVID-19 infection workers” (PCIW) in the study.

#### 2.2.3. COVID-19 Antigen and Antibody Tests

For the COVID-19 antigen tests, we used oligonucleotide primer RT-PCR for detection of the spike protein of SARS-CoV2 from nasopharyngeal swab specimens to confirm a recent COVID-19 infection. The results were reported as detectable, inconclusive or undetectable. The ELISA-based anti S1 protein IgG was used for the antibody test to indicate a remote COVID-19 infection, for which the cut-off values were classified as positive (>50 AU/mL), equivocal (3.8–50 AU/mL) or negative (<3.8 AU/mL). We used nasopharyngeal swabs to collect the samples for RT-PCR tests. At the time of this study, rapid antigen test kits (ATK) were not available in Thailand. All the suspected COVID-19 infection cases were tested with RT-PCR tests.

### 2.3. Data Collection and Analysis

We recorded general demographics, dormitory room number, workplace or departments in the factory, presence of respiratory symptoms at any time during the program, and each individual’s RT-PCR and/or COVID-19 antibody test result from randomly selected and mass testing. The obtained data were stored in a computerized database system operated by the Division of Data Innovation and Data Analysis (DIDA) of our faculty. We descriptively analyzed the associations between the collected parameters and the participants’ PCIS. Fisher’s exact test, Chi square or *t*-test was used to assess statistical significance (*p* < 0.05). The number of COVID-19 cases identified in the communities surrounding the factory during the time of the B&S program, as well as tracing of COVID-19 case contacts between the cases in the communities and the factory workers, was evaluated. We also described a timeline of the B&S program’s progress before achieving the percentage of PCIW required for the program exit.

### 2.4. Ethical Considerations

The study protocol was approved by the Ethics Committee of the Faculty of Medicine, Prince of Songkla University (EC code: REC.65-089-9-2; date 12 March 2022). We strictly complied with the 1964 Declaration of Helsinki and its related guidelines and amendments. The study participants’ identifiable information or personal data were completely anonymized. The data analysis was conducted in aggregation for protection of the study participants’ identities.

## 3. Results

### 3.1. Factory Overview

The factory had a total area of 18,000 m^2^, of which half was the factory warehouse and the rest included the production area, administration offices, engineering and machine unit, and worker dormitories. We enrolled 1539 LIW in this study. They worked in 10 departments of the factory, i.e., the pre-processing, production, warehouse, welfare office, quality control, quality assurance, human resources, factory administration, research and development, and engineering departments. The dormitory rooms for the LIW had a 4 × 4 m^2^ area containing an average of four workers in each room. For the LOW who travelled between their homes and the factory, “sealed transportation” and “sealed route” regulations were applied. “Sealed transportation” was achieved by separation of the rear seats of the factory pickup trucks into chambers using self-installed transparent plastic films on frames for an individual worker to sit in. “Sealed route” referred to the use of predefined and non-stop routes for transportation. During working hours, all the factory workers were required to work only in their workplaces without face-to-face contact with anyone from other areas. Moreover, they were encouraged to strictly comply with DMHTT measures, including avoiding close contact with other workers at meal-times, frequent hand washing, wearing a well-fitting face mask at all times when working and riding in the factory transportation trucks, regular body temperature measurements and reporting a respiratory illness to the factory health staff if one occurred. At the time of this study, 32 factories, including the study factory, were making various industrial products in Songkhla province. Significantly, no workers in any of these factories had received a dose of COVID-19 vaccine because of the limited availability of the vaccine around the country at that time.

### 3.2. Situation of COVID-19 Infection in the Factory and the Local Area

The first confirmed COVID-19 case in this factory was reported on 30 April 2021. Until 17 May 2021, 17 days after the first confirmed case was reported, the number of RT-PCR confirmed COVID-19 workers had cumulatively increased to 82 (2.8%), of which 33 (2.3%) were LOW, while 49 (3.2%) were LIW. This cumulative number of cases made an average of 4.8 COVID-19 cases diagnosed per day, or 14.5% (82/565) of workers who had respiratory symptoms (Figure 1). During the same month of April 2021, the number of new COVID-19 cases in Songkhla province was 629 (44.9:100,000), compared with 36,079 (54.5:100,000) cases in the country, which was the tenth highest number of provincial COVID-19 cases in Thailand at the time. Because the crowded living condition in the factory dormitories were considered as a major risk for speedy COVID-19 transmission, the factory administration team decided to start the B&S program on 21 May 2021 with the cooperation of the SPPHO and our faculty staff, and subsequently the “factory COVID-19 control committee” was formed.

### 3.3. Processing of B&S Program and Outcomes

At the beginning of the B&S program, the factory COVID-19 control committee conducted a triage among the LIW who had high COVID-19-related fatality risk using RT-PCR tests for COVID-19 detection. As the factory workers were young people with no underlying diseases, no workers were included in the triage. After the triage and before the COVID-19 antibodies tests were conducted, 60 of 62 LIW (96.8%) who had been identified as having respiratory illnesses during program progression had newly positive RT-PCR tests, which raised the infection rate to 3.9% (60/1539) in LIW. The last COVID-19 case in the LIW was reported in week 4 (16 June 2021) of the B&S program (Figure 2).

Nine days after the last COVID-19 case was diagnosed, the COVID-19 antibody tests were conducted in 133 randomly selected LIW for a preliminary estimation of the percentage of workers with PCIS. In cases with equivocal or negative COVID-19 antibody tests, two repeated RT-PCR tests 1 week apart were conducted to confirm or exclude infections. Here, we found 115 of the 133 (86.5%) LIW with PCIS.

To further evaluate the extent of positive PCIS among the study participants, a mass COVID-19 antibody testing was performed in the rest of enrolled LIW. In this evaluation, we found a total of 1278 positive antibody tests and 20 positive RT-PCR tests. In summation, there were 115 + 1278 + 20 = 1413/1539 workers or 91.8% of PCIW during the entire B&S program. Therefore, the committee decided to terminate the B&S program in this factory 9 weeks after the program initiation based on the percentage required for B&S program termination being exceeded (>85%) [12] (Figure 3). However, although the B&S program was ended, the personal preventive measures against COVID-19 transmission were continued under the supervision of the factory health team. After exiting the B&S program, there were no new COVID-19 outbreaks in this factory during a 1-year follow-up.

### 3.4. Spatiotemporal Association of Dormitory Rooms with PCIW

Only ‘female’ was significantly associated with PCIS in multivariate analysis. Three working areas of the factory had high numbers of infected cases, i.e., the production area, warehouse and human resources department (Table 1).

We tried to examine the spatiotemporal association between the location of the dormitory rooms in the dormitories with the time course of appearance of positive PCIS workers (or PCIW), but no associations were found. The positive PCIS workers presented randomly in all dormitory rooms along the time course during the B&S program without a definable pattern to indicate the originating center of the outbreak.

### 3.5. Impact of the B&S Program on the Surrounding Communities and Factory Productivity

During the B&S program in this factory, there were 30 new COVID-19 confirmed cases in the communities around the factory. There was no evidence of contact between these COVID-19 cases in the communities with the factory workers from the COVID-19 case contact tracings. Regarding the productivity of the factory, the factory production remained at nearly the same level as before the initiation of the B&S program, and all incoming orders from related enterprises were responded to without significant delay.

## 4. Discussion

The rapidly increasing number of the workers contracting COVID-19 in the study factory in early May 2021 alarmed the factory health team to start an active intervention to control COVID-19 transmission in the factory. Although the percentage of confirmed COVID-19 cases in all workers was only 2.8%, which was much lower than the percentage indicated for the start of a B&S program in a factory according to the MOPH-T’s guideline (10%), the COVID-19 control committee of the factory decided to initiate a “high intensity” B&S program promptly because it was considered that the crowded living conditions of the LIW in this factory were a significant risk factor for rapid COVID-19 transmission. The higher infection rate in LIW (3.2%) than in LOW (2.3%) supported this consideration. A previous study showed that >3 workers/dormitory room would increase the risk of SARS-CoV2 transmission [13]. However, we found no explainable pattern of COVID-19 transmission in the worker dormitories as no infection originating center or time course of the appearance of the positive PCIS workers could be demonstrated. Based on our findings, we suspect that, during the mealtimes where face masks were taken off, or other places, i.e., toilets, staircases, etc., where there were many shared contact surfaces and personal prevention measures were less monitored, these might have been the sources of the virus transmission in this factory. One study found that shared contact surfaces, especially toilet facilities in crowded buildings or public areas, were contaminated with SARS-CoV2 and sources of COVID-19 transmission [14]. Therefore, advice concerning scientifically based techniques including the types of disinfectant used and duration required for surface cleaning should be strictly followed.

By strictly following the practical guidelines of the B&S program, the factory health team successfully contained the COVID-19 virus in this factory within 9 weeks of the program’s initiation. The percentage of PCIW was as high as 91.8%, which was higher than required (>85%) for B&S program exit. We found that female was the only significant independent factor for acquiring COVID-19 during the B&S program in this factory. There was no clear explanation for this finding. We believe that this is possibly related to the socializing preference habits among the female workers (Table 1).

In Thailand, the B&S program was launched in August 2021 by the MOPH-T to control COVID-19 transmission among the workers in factories or other enterprises in which large numbers of workers were required, leading to crowded working and/or living conditions, and to prevent COVID-19 transmission to communities around the workplaces [12]. According to the recommendations of the MOPH-T, a B&S program should be started immediately by the health team of an enterprise in cooperation with the local health agencies whenever the percentage of COVID-19 cases reaches 10% or more in an enterprise. The interventions in the program were stratified into three levels of action intensity based on the COVID-19 infection rate in the enterprise. The criteria for selection of the action intensity of the program were as follows: (a) low intensity, when the percentage of COVID-19 cases was below 10%, (b) intermediate intensity, when the percentage of COVID-19 cases was 10% or more, and (c) high intensity, when the percentage of COVID-19 cases was more than 10% plus the total number of COVID-19 cases was more than 100, or new COVID-19 cases were reported on 14 days in any 28-day period [12]. In addition to the same practices as of the intermediate and low intensity interventions, the high intensity interventions included “factory isolation” of the COVID-19 workers or suspects instead of home or community isolation, and “sealed transportation and route” between the workers’ homes and the workplace. All the actions conducted in the program needed to be cooperatively monitored by the local governmental health agencies. Basically, a key principle of the B&S program was to group the workers working in the same workplaces or living in the same dormitory rooms without allowing personal contact across workplaces or dormitory rooms. To prevent COVID-19-associated deaths, the B&S program emphasized triaging workers who had high COVID-19 fatality risks by RT-PCR tests to receive treatment accordingly first. However, as most of the workers in this study were young people (age 20–40 years old) who generally had no underlying diseases, no workers in this factory were triaged to undergo RT-PCR test for COVID-19. Moreover, COVID-19 vaccinations for all high mortality risk workers and more than 70% of all workers were advocated by the program guideline [12]. As all the study workers were in good health condition, the lack of vaccinations in all workers was considered to be less likely to cause severe COVID-19 or dead among them, but the chance of rapid COVID-19 transmission was high, especially transmission from asymptomatic COVID-19 workers. Moreover, as ATK was not available at the study time, self-screening for COVID-19 among the workers was impossible. Our molecular microbiology staff put their maximal efforts to complete all the RT-PCR tests requested. These were concerns challenging the factory health team so that implementing an effective preventive protocol combined with close monitoring was necessary during the epidemic.

Four weeks after starting the B&S program, 60 (3.9% of LIW) newly RT-PCR-confirmed COVID-19 cases were reported making an infection rate of 2 cases/day, which was much lower than before program initiation. Thereafter, there were no additional new COVID-19 cases identified. Although the infection rate per day dropped to more than half of that before the B&S program, the percentage of positive RT-PCR tests in the workers with respiratory symptoms was markedly higher than that before the B&S program (95.2% vs. 14.5%). It was possible that minor or asymptomatic COVID-19 workers in our study were under-reported before starting the B&S program. A COVID-19 anti-S1 IgG antibody study in 392 COVID-19 convalescents reported 8.7% of asymptomatic cases [15]. However, spreading of the disease to the local communities from the factory workers was not found as 30 new COVID-19 cases in the communities around the factory were not found to have contact with COVID-19 factory workers. Hence, it can be concluded that the program was successful in containing the disease in the workplace and preventing contamination to the surrounding communities.

One study which investigated the presence of COVID-19 antibodies in post-COVID-19 infected persons found that Fc-dependent antibody activity in mild to moderate COVID-19 infections persisted longer than neutralization antibodies after a SARS-CoV-2 infection [16]. Other studies have reported that antibody activity against COVID-19 re-infection persisted for 4–14 months after the index COVID-19 infection [15,17,18,19,20,21,22,23,24]. This could explain no further outbreaks 1 year after the program was ended in this factory. Our findings indicate that a B&S program can be effective in the prevention of COVID-19 re-infections, especially in unvaccinated workers as our study participants.

Since it is desirable for businesses to continue for maintaining financial sustainability, the shortage of manpower resulting from avoidance or prohibition of human gatherings due to the concern of disease transmission was a problem. The longer the time a business was disrupted or eventually shut down due to COVID-19 infection control, the larger the economic loss ensued in return. While complete lockdown measures were previously employed to prevent human gatherings aimed at controlling viral transmission, mobilization of the workers through several transportation routes back to their hometowns created a risk of dispersing the disease in their local communities, as occurred during the first wave in Thailand earlier in the year 2020. Therefore, the B&S program was devised and implemented to balance the two opposing arms, i.e., sustainability of business and controlling COVID-19 transmission. In other words, it was believed that the B&S program was able to mitigate both COVID-19-related health harms and economic crises by allowing businesses to be operated as close as possible to the usual capacity before the epidemic.

Our study described the process, outcome and impact of a B&S program for COVID-19 transmission control in a factory in southern Thailand, where LIW were considered as a high-risk group for disease transmission. Although this was a single factory experience, we believe that the details of the program will be useful to apply to other workplaces with similar settings as ours. Many studies have supported absolute lockdown and mass screening for COVID-19 for disease transmission control. Studies from Kuwait and Singapore, where complete lockdowns of migrant worker dormitories in the factories were applied, reported effective disease control [8,9]. Mass COVID-19 screening and migrant worker dormitory lockdown measures were also applied during the epidemic in Samutsakorn province and other worksites in Thailand as well [1]. The B&S program in this study was not an absolute lockdown protocol since about half of the workers were LOW who possibly had contact with other people in their homes or communities. However, it was possible that, under the strict factory regulations and supervision for COVID-19 transmission control, combined with the public reinforced recommendations for personal prevention, these could act together to strengthen disease transmission control. One study supported the findings that encouraging factory workers’ compliance with hand washing and physical distancing was significant in controlling COVID-19 transmission [25]. Another study reported that multiple social determinants, such as factory policies about working and living conditions, workers’ socioeconomic constraints, the accessibility of health care, especially a complete vaccination program, and reliable health information provided by official public health organizations, influenced the cooperative adoption of personal prevention measures among migrant workers [26,27]. Hence, these factors should be thoroughly considered in drafting and shaping local health strategic policies to control the disease epidemics. Particularly, migrant labor workers should be a group of people receiving a special concern as they are usually associated with inferior social determinant factors such as low education, low socioeconomic status and unequal access to standard medical care. These factors jointly cause them to be highly vulnerable people for contracting and spreading COVID-19 unintentionally. We believe that the success of the B&S program in this study was due mostly to integration of all social determinants mentioned in planning for COVID-19 transmission control. A similar study to ours recommended an integrative total worker health framework (TWHF) for helping employers to organize a system to protect their workers’ health, increase public health security against disease transmission and to sustain business activities. The TWHF consisted of six key characteristics as follows: (1) improve working condition; (2) utilizing a participatory approach; (3) employing comprehensive and collaborative efforts; (4) commitment as a leader; (5) adherence to ethical and legal standards; and (6) using data to guide actions [28]. The recent COVID-19 pandemic had different characteristics from the earlier ones. Not only did the epicenter of the disease change from entertainment venues to markets and other workplaces, but also the involved population changed from the aged with underlying comorbidities leading to high fatality risk, to young and healthy workers with only mild or asymptomatic disease [8,9,10,29]. Under the situation of inaccessible to mass vaccinations for migrant workers, multimodal and integrative personal and societal prevention measures as mentioned earlier, not limited to mass vaccinations only, had been suggested for disease control [30]. We thought that the best practices we had learnt in facing this crisis were: (1) early detection of an alarming sign of COVID-19 outbreak in an enterprise, and then triage the cases with high mortality risk, isolate and treat them; (2) application of appropriate measures balancing workers’ and local community people’s health safety and business sustainability; (3) clearly inform and ask for compliance from the workers by stressing on the disease harm and benefits of compliance to the health recommendations; and (4) respect the workers’ human rights. Although this protocol seemed to restrict the workers’ individual freedom, the factory health team respected their basic human rights as the essential materials for daily living were fully supplied without charge, including their regular wages despite no working. They were allowed to leave the factory if they had a medical emergency beyond the facilities of the factory medical care, or any suitable reasons presented themselves. Additionally, one or more translators who understood the workers’ native language were provided to help the communications between the workers and the factory health team staff or the other persons they required contact with. This aimed to reduce the cultural or social gaps between the workers and factory health team staff, which possibly led to the workers’ psychological stress.

The strength of the current study was the description of a real situation of a B&S program implementation, including program outcomes and public impact. The data obtained were systematically stored in a computerized database at our faculty, which was feasible for real-time analysis. The limitation of this study was that it was a single site study and no comparison with other factories in the same area was conducted. We suggest that comparisons of COVID-19 infection rates between the LIW and LOW should be performed to evaluate the overall usefulness of the program, which used a partially sealed protocol. Additionally, we encourage a comparative study between workplaces where B&S programs are applied and those where they are not in the future as well.

## 5. Conclusions

COVID-19 brought about unprecedented worldwide economic losses besides the health crisis. Regulating physical distancing and quarantining eventually led to widespread disruption of industrial productivity and business activities. As overcrowded working or living conditions in a workplace is a known risk of speedy viral transmission, mandating and monitoring of the practical protocols included in the B&S program, and emphasizing workers’ strict compliance to personal hygiene guidelines, are the keys to success, even though a partially sealed protocol of the B&S program is applied, as in this study. We hope that the learning points gained here will be useful in preparation for the next COVID-19 or other infectious disease epidemics, if they occur, and future research questions regarding strategies for COVID-19 epidemic control in workplaces.

## Figures and Tables

**Figure 1 ijerph-19-16391-f001:**
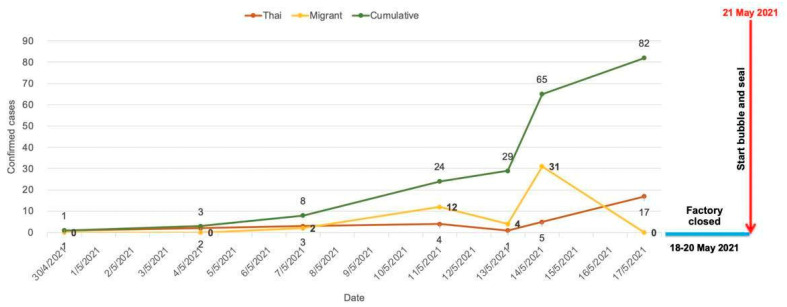
Date of diagnosis and number of RT-PCT-confirmed COVID-19 cases before starting the Bubble and Seal program.

**Figure 2 ijerph-19-16391-f002:**
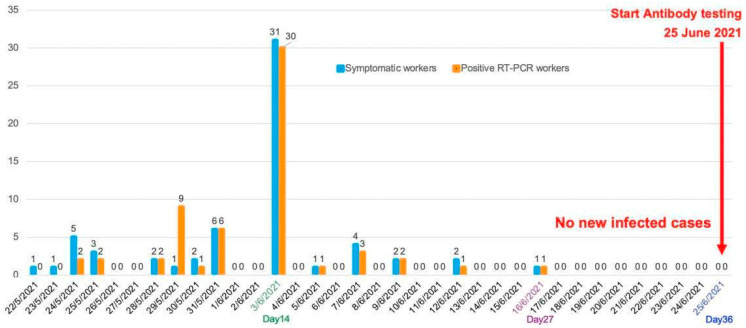
Date of diagnosis and number of RT-PCR confirmed COVID-19 workers among the migrant workers with respiratory symptoms during the time between the triage and antibody testing of the Bubble and Seal program (21 May 2021–25 June 2021).

**Figure 3 ijerph-19-16391-f003:**
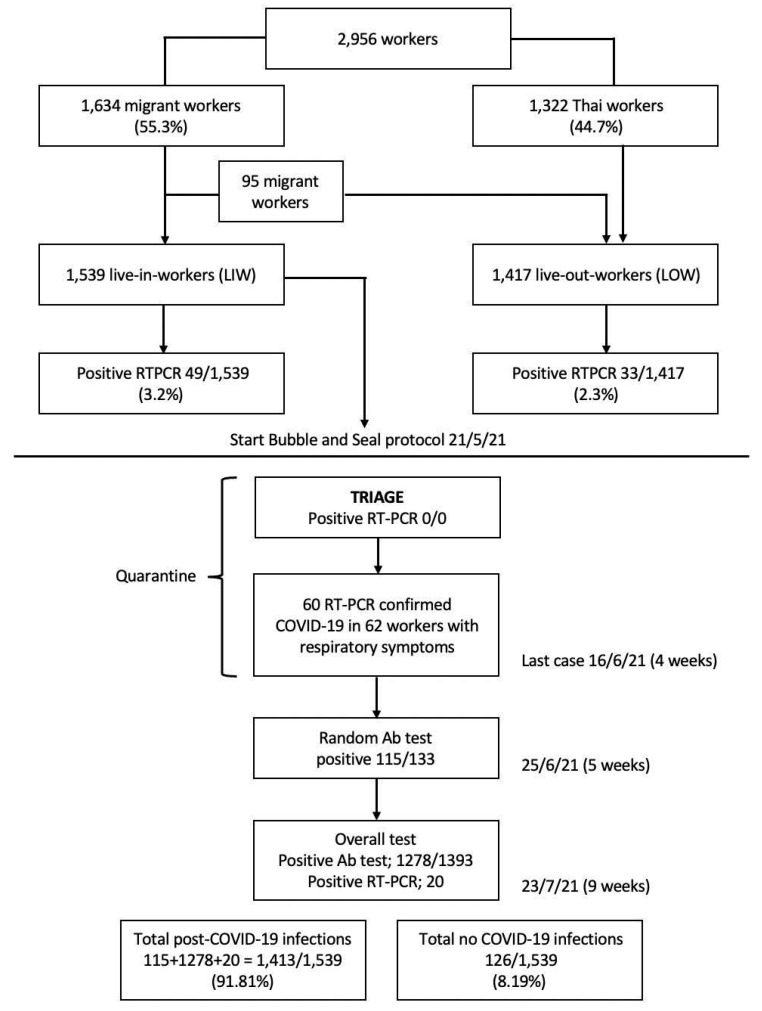
Flow chart of the COVID-19 situation of the factory before and during the Bubble and Seal program.

**Table 1 ijerph-19-16391-t001:** Study variables which were associated with positive post-COVID-19 infection status (PCIS) (positive RT-PCT for COVID-19 or COVID-19 antibody test) (n = 1533, 6 cases missing).

Variable	PCIS +	PCIS −	*p*-Value
**Sex, n (%)**			
-Male	607 (87.97)	83 (12.03)	*p* = 0.000 * (Chi square = 29.645)
-Female	806 (95.61)	37 (4.39)	
**Age group (years)**			
≤20	11	2	*p* = 0.499 (Fisher’s exact)
21–30	773	59	
31–40	531	51	
41–50	91	8	
51–60	7	0	
**Department, n (%) [—ok to leave out ‘department’ here, understood]**			
Pre-processing area	6	2 (25)	*p* = 0.061 (Fisher’s exact)
Warehouse	91	8 (8)	
Quality control	3	0 (0)	
Human resources	38	2 (5)	
Administration	0	2 (100)	
Quality assurance	1	0 (0)	
Production area	1241	104 (7)	
Welfare office	2	0 (0)	
Research and development	10	0 (0)	
Engineering	21	2 (8)	
**Dormitories**			
A	518	37	*p* = 0.170 (Chi square = 3.546)
B	644	32	
C	285	23	

* *p* < 0.05, Chi square test.

## Data Availability

The study data and analysis methods were described in the materials and methods section. No data were deposited in other pre-print servers.

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
