# Peer review of "Integrative Effects between a Bubble and Seal Program and Workers’ Compliance to Health Advice on Successful COVID-19 Transmission Control in a Factory in Southern Thailand"

_ijerph, 2022, doi:10.3390/ijerph192416391_

Round 1

Reviewer 1 Report

Overall, the manuscript is interesting to read. However, the significance of the study should be brought out clearly. The author(s) should clearly include how the findings contribute to the theory, practice, and research. they may further discuss the best practices so that others can adopt them when facing a similar crisis in the future. How are the findings important to crisis management research?

Reviewer 2 Report

I read the proposed article with great interest. The authors report their experience with a Bubble and Seal protocol to control the spread of COVID -19 infection in a seafood production facility. This is undoubtedly an approach that has yielded benefits, as can be seen from the results proposed by the authors. A balance was found between protecting public health and not bringing economic productivity to a complete standstill. From this point of view, it was a remarkable effort that can certainly inspire others. However, some ambiguities remain that I would ask the authors to clarify.

- The aspect of whether or not the workers voluntarily participated in this project is never addressed. Were they adequately informed? Did they give their consent to participate in this project? This meant a considerable restriction of their individual freedom. It would be very appropriate if the authors clarified this aspect not only in the materials and methods but also in the discussions. In addition, many of the participating workers were migrants: were there cultural and social problems in these cases? 

- The authors do not specify what kind of test was conducted. I assume it was nasopharyngeal swabs, is that correct? It would also be interesting to reflect in the discussions about the need for minimally invasive testing in patient health surveillance programs. Then the use of both antigen and RT-PCR tests would be explained 

These are the two main aspects that would merit further consideration.
